# Azithromycin consumption during the COVID-19 pandemic in Croatia, 2020

**Nikolina Bogdanić**[1]*, **Loris Močibob**[1], **Toni Vidović**[2], **Ana Soldo**[2], **Josip Begovać**[1,3]

**1** University Hospital for Infectious Diseases, Zagreb, Croatia, **2** Zagreb City Pharmacy, Zagreb, Croatia, **3** School of Medicine, University of Zagreb, Zagreb, Croatia

* nikolinabogdanic@gmail.com

**Data Availability Statement:** We shared the data underlying the results on public repository recommended by PLOS. URL: https://mfr.osf.io/render?url=https%3A%2F%2Fosf.io%2Fp479v%2Fdownload.

## Abstract

### Background

During the initial phase of the COVID-19 pandemic, there was great enthusiasm for the use of azithromycin with or without hydroxychloroquine.

### Objectives

We analyzed azithromycin consumption in Croatia in 2020 and compared this to the period 2017–2019.

### Methods

Azithromycin consumption was evaluated using the IQVIA Adriatic d.o.o. database which collects data on azithromycin distribution from wholesale pharmacies to hospital and non-hospital pharmacies in Croatia. We analyzed data for the period from January 2017 to December 2020. Azithromycin distribution was measured as days of therapy (DOT) and reported as per 1000 inhabitants or per 1000 inhabitant-days.

### Results

In the period 2017–2020, total azithromycin DOT in Croatia increased in 2017, 2018, 2019, and 2020 (1.76, 1.91, 1.91 and 2.01/1000 inhabitant-days, respectively). Non-hospital pharmacies received 2.18 times and hospital pharmacies 4.39 times more DOT units/1000 inhabitants of azithromycin in March 2020 compared to the average distribution rate in March 2017–2019. During the peak of the COVID-19 epidemic (November and December 2020) azithromycin distribution increased considerably in hospital (3.62 and 3.19 times, respectively) and non-hospital pharmacies (1.93 and 1.84 times, respectively) compared to the average consumption in the same months in 2017–2019.

### Conclusions

Our data showed increased azithromycin distribution in the period 2017–2020 which indicates azithromycin overuse. Preliminary information on COVID-19 treatments with a desire to offer and try what is available even in the absence of strong scientific evidence may have influenced practices of antimicrobial prescriptions.

**Funding:** The funders had no role in study design, data collection and analysis, decision to publish, or preparation of the manuscript. The authors received no specific funding for this work.

**Competing interests:** This commercial affiliation does not alter our adherence to PLOS ONE policies on sharing data and materials.

## Introduction

Since the beginning of the COVID-19 pandemic concerns about the potential consequences of antimicrobial overuse have been raised [1]. At first, increased patient exposure to antimicrobials was caused by the lack of rapid diagnostic tools, decision tools, and concerns of bacterial coinfection [1]. An international survey among physicians involved in the treatment of COVID-19 patients on antibiotic prescribing practices conducted in April 2020 revealed that the decision on antibiotic use was mostly based on clinical presentation, with the need for coverage of atypical pathogens and more than half of the participants reported use of a combination of β-lactams and macrolides or fluoroquinolones [2].

Azithromycin has antibacterial activity, an immunomodulating effect, and perhaps even some antiviral activity [3]. The use of azithromycin with hydroxychloroquine gained much attention when preliminary observational reports and non-randomized pilot studies indicated favorable outcomes in patients treated with this combination [4, 5]. *In vitro* experiments suggested a synergistic effect with hydroxychloroquine on inhibition of viral replication [6].

Preliminary information on COVID-19 treatments with a desire to offer and try what is available even in the absence of strong scientific evidence might have influenced practices of antimicrobial prescriptions. We investigated azithromycin consumption in Croatia in 2020 and compared this to the period 2017 to 2019.

## Materials and methods

The data on azithromycin distribution to hospital and non-hospital pharmacies in Croatia (population 4.058 million in 2020) served as a proxy to consumption and were extracted from the electronic database of IQVIA Adriatic d.o.o. IQVIA Adriatic d.o.o. collects data on the quantity of individual drugs distributed from the wholesale pharmacies to all retail and hospital pharmacies in Croatia.

We analyzed data on azithromycin distribution for the period from 1$^{st}$ January 2017 to 31$^{st}$ December 2020. Azithromycin consumption was measured as days of therapy (DOT) which represents a ratio of the total dose of azithromycin in one package divided by the defined daily dose unit of 0.3 gr. We present our main data in absolute numbers and those based on 1000 inhabitants-days of azithromycin distribution. The monthly DOTs (hospital and non-hospital) are reported per 1000 inhabitants and presented graphically. The comparisons between years are reported as percent or rate difference.

The data on population size, based on the yearly national population, was extracted from Eurostat online platform. To compare DOT between years we used the ratio of the monthly rate in 2020 and the average corresponding monthly rate of the years 2017–2019.

Spearman's rank-order correlation test was used to measure the association between quantities of azithromycin distribution and the number of COVID-19 cases (Fig A in S1 File). To formally compare the distribution of azithromycin to pharmacies between 2020 and the average of 2017 to 2019 we used a 2-tailed *t*-test for independent samples. The comparison was made by the mean monthly distribution for each quarter, and for this analysis, we used a unit of azithromycin of 1500 mg/day per 1000 inhabitants. A unit of azithromycin of 1500 mg/day is a standard course of treatment and could be a proxy for one prescription. Results are shown in Table A in S1 File. Statistical significance was set at P<0.05. The statistical analysis was done with SAS software, version 9.4. (SAS Institute, Cary, NC, USA) and graphs were ploted using GraphPad Prism version 9.3.1 for Windows, GraphPad Software, San Diego, California USA. The study was approved by the Ethical Committee of the University Hospital for Infectious Diseases, Zagreb, Croatia.

## Results

In the period from 2017 to 2020, total azithromycin DOT distribution per calendar year increased in 2017, 2018, 2019, and 2020 (1.76, 1.91, 1.91, and 2.01/1000 inhabitant-days, respectively). This was a 5.2% increase from 2019 to 2020 and an 8.1% increase from the average of 2017–2019 to 2020. This 8.1% increase corresponds to a total of 37224 5-day courses of azithromycin prescriptions. Azithromycin was distributed to non-hospital pharmacies in 93.2% to 95.6% of total DOT, with an increasing trend of distribution to hospital pharmacies from 2017–2020 (4.4, 4.5, 5.1, and 6.8% of total DOT, respectively). The total annual amount of azithromycin DOT units distributed was lowest in 2017 (2.670 million units) and highest in 2020 (2.976 million units), whereas the population adjusted distribution was 642.85/1000 inhabitants in 2017 and 733.43/1000 inhabitants in 2020.

The monthly pattern of azithromycin distribution was quite different in 2020 compared to the previous three years (Fig 1). Azithromycin distribution to both hospital and non-hospital pharmacies in 2020 was highest in March, followed by November and December. The number of COVID-19 cases in Croatia started to increase exponentially from September 2020 and the epidemic reached its peak in November and December 2020 with 79126 and 82395 cases, respectively (Fig 1). There was a positive correlation between the number of COVID-19 cases and the total azithromycin consumption from July to December 2020 (Spearman's test, $\rho$ = 0.94, p = 0.005, Fig A in S1 File). Among hospital pharmacies, azithromycin DOT in March 2020 was 4.39 times higher in comparison to average azithromycin DOT in March 2017–2019

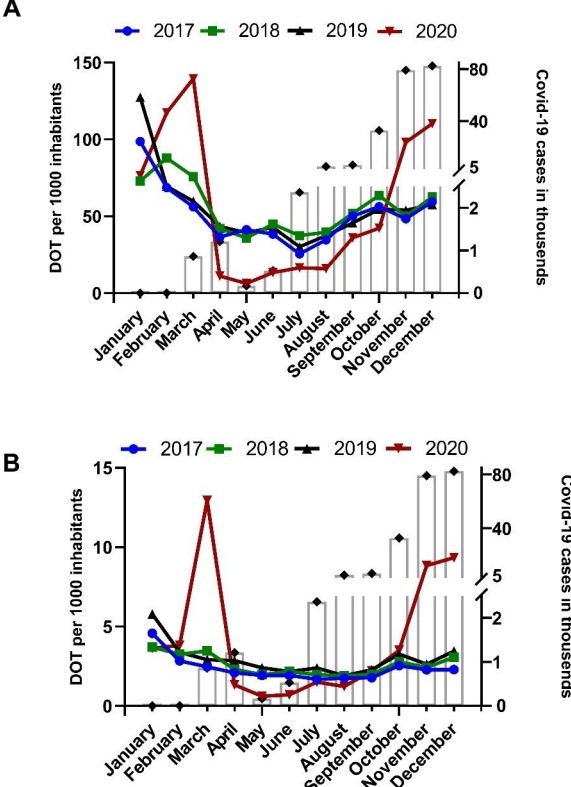

**Fig 1.** Days of therapy (DOT) units of azithromycin dispensed per month to non-hospital (panel A) and hospital (panel B) pharmacies in Croatia from 2017 to 2020. Open bars are number of COVID-19 cases. A strict lockdown was in place from March 19[th] to April 27[th], 2020.

(12.95/1000 inhabitants *vs.* 2.95/1000 inhabitants), 3.62 and 3.19 times higher in November and December 2020 (8.84/1000 inhabitants *vs.* 2.44/1000 inhabitants and 9.34/1000 inhabitants *vs.* 2.93/1000 inhabitants), respectively (Fig 2). Among non-hospital pharmacies, azithromycin distribution in March 2020 was 2.18 times higher in comparison to the average March azithromycin distribution in 2017–2019 (DOT, 139.51/1000 inhabitants *vs.* 63.94/1000

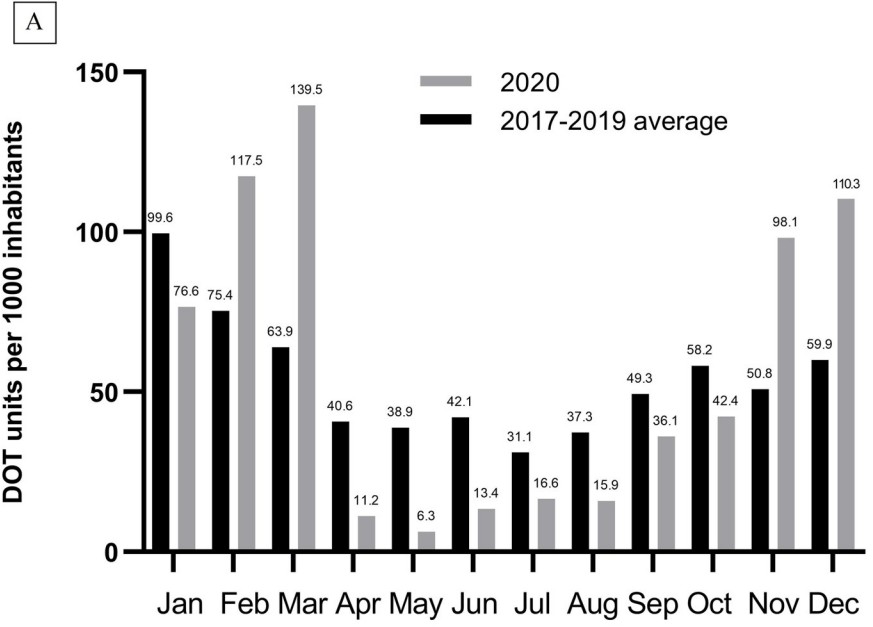

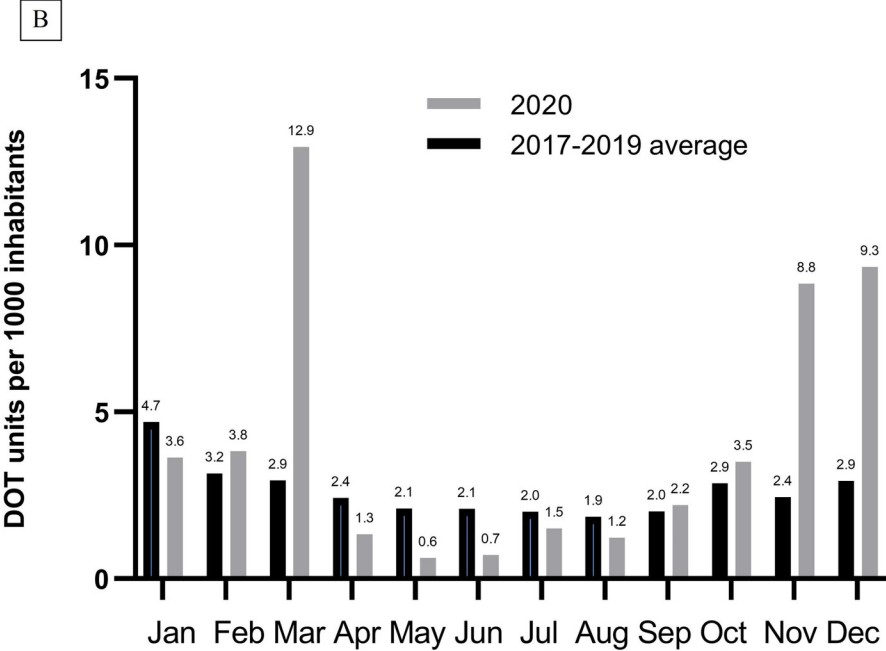

**Fig 2.** The monthly distribution of azithromycin in 2020 compared to the average of 2017 to 2019 for non-hospital (panel A) and hospital (panel B) pharmacies in Croatia. DOT, days of therapy.

inhabitants), 1.93 and 1.84 times higher in November and December 2020 (DOT, 98.15/1000 inhabitants *vs*. 50.83/1000 inhabitants and 110.28/1000 inhabitants *vs*. 59.92/1000 inhabitants), respectively (Fig 2). In April-August 2020 the distributed DOT of azithromycin was not only lower than in other months of 2020, but also lower than the average in April-August 2017–2019 for both hospital and non-hospital pharmacies (Fig 2). In fact, the distributed DOT of azithromycin in April-June 2020 was statistically significantly lower compared to the average of 2017–2019 (Table A in S1 File).

## Discussion

Azithromycin consumption increased in the period from 2017 to 2020 and a total of extra 37224 standard courses (1500 mg) of azithromycin treatment were dispensed in 2020 compared to the average of 2017–2019. The pattern of azithromycin monthly distribution in 2020 was different in comparison to previous years, probably due to various reasons. Azithromycin distribution to hospital and retail pharmacies was largest in March when the first small study of French authors on its efficacy in COVID-19 was published on a preprint server [7]. Also, the era of social media and its high impact on people's perception of disease-related themes created an infodemic. Galotti et al. analyzed Twitter messages posted from January to March 2020, and found an increase in potentially unreliable information in those early stages of the COVID-19 pandemic [8]. Niburski et al. analyzed Amazon purchases of azithromycin which increased in March 2020 after US President Donald Trump endorsed its use in social media posts and public speeches [9]. It thus seems that fear of a great epidemic and uncertainty due to the absence of the treatment with proven efficacy can influence consumption of antimicrobials with assumed or unproven benefits. However, US data on the number of patients dispensed antibiotics from retail pharmacies in March 2020 was similar to that in 2017–2019 [10]. In our study, it seems that the huge supplies acquired by hospitals and retail pharmacies in March 2020 have been dispensed to patients over the following months because the distribution of azithromycin was several times lower in the period from April to August 2020 compared to 2017–2019 (Fig 2). Data from the US showed that the number of patients dispensed azithromycin in May 2020 was 62% lower than the historic average, and this was assumed to be related to mitigation measures undertaken to curb the COVID-19 epidemic [10]. Sluggett et al. reported that monthly rates of distributed macrolides in outpatient settings in Australia were lower in October 2019 to September 2020 when compared to October 2018 to September 2019 for every month except March 2020 when it was 19.3% higher [11].

The monthly pattern of azithromycin distribution was similar in 2017, 2018 and 2019 with higher distribution during winter months. It is known that every influenza season leads to increased use of antibiotics in the winter months [12]. Isolation measures and social distancing during 2020 probably contributed to fewer acute respiratory infections including post-influenza pneumococcal infection and atypical pneumonia which is a common reason for azithromycin use during the influenza season. In the 2020/2021 season the influenza epidemic was at a very low level in the whole European Region and it could not have had a significant impact on antimicrobials prescription in Croatia in 2020 [13]. So, the facts that there was no significant influenza epidemic would indicate that the increase in azithromycin distribution was mainly due to COVID-19 and suggests that the true overconsumption was higher than the 8.1% increase found in our study.

At the beginning of the COVID-19 pandemic, there were concerns about bacterial coinfections in COVID-19 patients. In a review by Langford et al. antibiotics were prescribed in 74.5% (95% CI 68.3–80.0%) of 30 623 patients while estimate of bacterial coinfection was 8.6% (95% CI 4.7–15.2%) [14]. Rawson et al. also reported that only 8% of hospitalized patients had

confirmed bacterial or fungal coinfection [15]. Coinfection with *Mycoplasma pneumoniae* and *Chlamydia pneumoniae* might justify the use of azithromycin in COVID-19. A study from North California which analyzed 161 nasopharyngeal swab specimens of COVID-19 patients did not show coinfections with *Mycoplasma pneumoniae* and *Chlamydia pneumoniae* detected by RT-PCR [16]. Higher rates of coinfections were found in studies using serologic tests for the diagnosis of *Mycoplasma pneumoniae* and *Chlamydia pneumoniae*; in a retrospective Italian study 242 of 434 hospitalized COVID-19 patients were coinfected with *C. pneumoniae* and/or *M. pneumoniae* [17]. However, PCR-based assays are preferred for diagnosis of both *C. pneumoniae* and *M. pneumoniae* while serologic testing makes a retrospective diagnosis of infection with an uncertain clinical significance in COVID-19. In summary, the data on the role of *C. pneumoniae* and *M. pneumoniae* coinfection in COVID-19 is still unclear and currently does not support the routine use of azithromycin in COVID-19.

Azithromycin consumption gradually increased from July and peaked in November and December 2020 (Figs 1 and 2). In September 2020 the results of a randomized study showed no benefit on clinical outcome of adding azithromycin to standard of care treatment and this has been confirmed in the RECOVERY randomized trial [18, 19]. Despite these findings and low occurrence of bacterial coinfections [15], azithromycin distribution to hospital pharmacies increased about 3 times in November and December 2020 in comparison to the average of the same months in 2017–2019. The increase was less pronounced in non-hospital pharmacies; however, it was almost 2 times higher when compared to 2017–2019 (Fig 2).

Our study has limitations. The data on individual patients were not available and we could not follow up dispensed azithromycin from pharmacies to patients. However, it is quite certain that all quantities of distributed azithromycin have been dispensed to patients. We were also not able to report more standard measures of antibiotic usage such as defined daily dose or DOT per 1000 patient-days. Our formal analysis looking at quarterly distribution of azithromycin did not show a statistical significance between the mean monthly distribution for the 1st and 4th quarter in the period 2017 to 2019 compared to 2020. However, this analysis has limited power to detect a difference. Furthermore, this result should not be interpreted as if overconsumption of azithromycin did not occur in 2020. The amount of extra azithromycin distributed in 2020 compared to previous years is equivalent to almost 38 000 extra 5-day courses with no obvious other reasons but COVID-19 and this cannot be ignored. One may also speculate about other non-COVID-19 causes of azithromycin overconsumption in 2020. However, this is very unlikely as there was no report from the Croatian National Public Health Institute of any other ongoing epidemic requiring azithromycin treatment in 2020.

Even though an increase of 8.1% of total azithromycin DOT from the average of 2017–2019 compared to 2020 seems modest, we should highlight the fact that Croatia is already one of the countries with highest percentage of macrolide resistant isolates among EU/EEA countries [20]. Macrolides were the second most common used antibiotics with azithromycin accounting for most of the use in Croatia in 2014–2019 with daily defined doses (DDD)/1000 inhabitants per day of 2.58, 2.82, 2.55, 2.56, 2.73, 2.72, respectively, after penicillins including beta-lactamase inhibitors [21, 22]. Sample of *Streptococcus pneumoniae* with estimated national population coverage of 80% showed increasing trend in macrolides resistance in the period 2015–2018 (19, 34, 36, 32%, respectively) [20]. Resistance data collected from 38 centers in Croatia in 2019 showed 31% resistant *S. pneumoniae* to macrolides with a range of local results from 0 to 53% and 9% of macrolide resistant *Streptococcus pyogenes* with a range of local results from 0 to 25% [23]. While widespread use of antimicrobials leads to more bacterial resistance which is of great public health concern, it also exposes patients to possible side effects. Azithromycin is generally a well-tolerated and safe drug, however, when combined

with other QT-prolonging drugs (such as hydroxychloroquine) it may increase the risk for serious toxicities and requires careful monitoring.

In conclusion, we provided national data on overuse of azithromycin during the COVID-19 epidemic in Croatia in 2020. Antibiotic stewardship principles should not vanish during the challenging times of the COVID-19 pandemic. To preserve the effectiveness of existing antimicrobials such as azithromycin, antibiotic stewardship principles should be followed and carefully applied in times of crisis.

## Supporting information

**S1 File. Correlation analysis of COVID-19 cases and azithromycin days of therapy and comparison of the mean monthly distribution of azithromycin by quarters between 2020 and the average of 2017 to 2019.**
(DOCX)

## Acknowledgments

We thank IQVIA Adriatic d.o.o. for providing the data on azithromycin distribution.

## Author Contributions

**Conceptualization:** Nikolina Bogdanić.

**Data curation:** Loris Močibob, Toni Vidović.

**Formal analysis:** Josip Begovać.

**Investigation:** Ana Soldo.

**Methodology:** Nikolina Bogdanić.

**Supervision:** Josip Begovać.

**Writing – original draft:** Nikolina Bogdanić, Josip Begovać.

**Writing – review & editing:** Loris Močibob, Toni Vidović, Ana Soldo.

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
