## [Decision Letter · Decision Letter 0]

26 May 2021

PONE-D-21-11999

Azithromycin consumption during the COVID-19 pandemic in Croatia, 2020

PLOS ONE

Dear Dr. Bogdanic,

Thank you for submitting your manuscript to PLOS ONE. After careful consideration, we feel that it has merit but does not fully meet PLOS ONE’s publication criteria as it currently stands. Therefore, we invite you to submit a revised version of the manuscript that addresses the points raised during the review process.

Please make necessary changes as per reviewer comment. And submit accordingly.

We look forward to receiving your revised manuscript.

Kind regards,

Prasenjit Mitra, MD, MRSB, MIScT, FLS, FACSc, FAACC

Academic Editor

PLOS ONE

Journal Requirements:

4.Thank you for stating the following in the Financial Disclosure section:

We note that one or more of the authors are employed by a commercial company: Zagreb City Pharmacy

Reviewers' comments:

Reviewer's Responses to Questions

**Comments to the Author**

1. Is the manuscript technically sound, and do the data support the conclusions?

Reviewer #1: Yes

2. Has the statistical analysis been performed appropriately and rigorously? 

Reviewer #1: Yes

3. Have the authors made all data underlying the findings in their manuscript fully available?

Reviewer #1: Yes

4. Is the manuscript presented in an intelligible fashion and written in standard English?

Reviewer #1: Yes

5. Review Comments to the Author

Reviewer #1: I want to compliment the authors. This is a well written manuscript, giving very interesting information, with lessons to be learned for the future. I have some remarks/suggestions for the authors to consider, that I hope could further strengthen their case.

Abstract:

- from 2017 to 2020 (1.76, 1.91, 1.91 and 2.01/1000 inhabitant-days, respectively)

suggestion for clarity: increased in 2017, 2018, 2019 and 2020

- Azithromycin distribution to pharmacies was the largest in March 2020 which indicates that fear of a great epidemic can influence consumption of antimicrobials with unproven benefits for the disease in question. Our data indicate that azithromycin overuse was also present during the second wave of the COVID-19 epidemic in Croatia.

I think the DOT over the years is the best measure of AZ overconsumption, as the monthly DOT are difficult to interpret. Distribution is, as the authors correctly state in the limitations, not the same as use; and we can assume that with the quick succession of scientific reports on this issue, it will have been very difficult to attune supply and demand. While I agree that the monthly evolution is described in the abstract, I would personally leave it out of the conclusion.

I think it is rather bold to assume that fear has been the main drive for azithromycin consumption. Also, until October, there were to my knowledge still only two randomized controlled trials (Cavalcanti et al, Furtado et al), in which azithromycin was mostly combined with hydroxychloroquine (overview doi: 10.1136/bmjresp-2020-000806). I agree that AZ should have ideally only been prescribed in a context of randomized controlled trials. However, with the conflicting data (and no direct safety issues reported with AZ monotherapy) I can understand why hospitals with more limited resources (in terms of dedicated study personnel) would have chosen to use AZ outside of studies until the results of RECOVERY were widely known. I think this sentence “ Preliminary information on COVID-19 treatments with a desire to offer and try what is available even in the absence of strong scientific evidence might have influenced practices of antimicrobial prescriptions.” Is much stronger, closer to the truth and suits the conclusion better. I would rather reform the current sentence in the conclusion as a question for the introduction, e.g. “could fear of a great epidemic influence consumption of antimicrobials with unproven benefits?”, or else raise this issue in the discussion.

- About the 8.1% increase compared to the mean of 2017-2019

o Compared to 2018-2019 the increase is only 5%. I have no view on AZ prescription in Croatia, but I feel their case would be stronger if AZ prescription over the last 5 years was mentioned instead of only the last 3. Given that this makes the figure more complex, I would understand why the authors would stick to 3 years, but I would kindly ask if they could provide me those numbers then.

o I presume that in Croatia too, a significant part of AZ prescription will be chronic AZ use (e.g. 3 weekly doses for COPD), which of course greatly influences the DOT/1000 inhabitant days. I wonder if the average reader will be aware of the fact that an 8,1% increase is therefore more important than it looks at first sight, because treatment duration for COVID is shorter. An absolute increase of 0.15DOT/1000inhabitant days is quite abstract, but in a population of approximately 4000000inhabitants, and given that a normal AZ treatment course for COVID would be 5 days, I’d say that about an extra 43000 patients (!!) were treated with AZ. I would even suspect that the true overconsumption due to COVID has been larger. One should subtract from the mean of 2017-2019 also the number of cases that have not been treated from atypical pneumonia in the Influenza season, which was considerably less than previous years due to isolation measures. The latter is marginally mentioned in the manuscript but could be emphasized more to underline the magnitude of the increase.

- AZ monotherapy seems safe from the available data. However, combination with HQ or with other antibiotics on ICU raises concerns for cardiac adverse events and warrants close monitoring. This could be emphasized in the discussion and related to the significant increase in (often frail) patients that have been exposed to the drug.

- At the other hand, for the authors, the most important concern on the long-term is probably microbial resistance against macrolides. For this I personally think that it is reassuring that the relative increase has ONLY been 8%. I think that the discussion would be stronger if the authors could elaborate on that.

o E.g. do they think that this 8% increase in DOT, which will have mostly been spread over short treatment courses (causing less resistance than chronic treatment) of approximately 5 days, will have a significant impact on bacterial resistance? To the best of my knowledge, there is no data for my own country yet… is ther for Croatia?

o Are we lucky that COVID mitigation strategies caused an unusually mild influenza season (influenza having a notoriously larger number of bacterial coinfections than COVID-19), which has somewhat compensated for the ungrounded increase of AZ in the treatment of COVID?

I have no additional comments

Kind regards

6. PLOS authors have the option to publish the peer review history of their article (what does this mean?). If published, this will include your full peer review and any attached files.

Reviewer #1: **Yes: **Iwein Gyselinck

---

## [Author Response · Author response to Decision Letter 0]

19 Jun 2021

We are grateful to the reviewer for the insightful comments on our paper. We have incorporated most of the suggestions made by the reviewer and made necessary changes. Enclosed herein is the revised version of our manuscript. Please see below, in italics, for a point-by-point response to the editor’s and reviewer’s comments. 

° Please ensure that your manuscript meets PLOS ONE's style requirements, including those for file naming.

We checked PLOS ONE’s style requirements and made necessary changes to our manuscript. For details please see revised manuscript.

° Please review your reference list to ensure that it is complete and correct. If you have cited papers that have been retracted, please include the rationale for doing so in the manuscript text, or remove these references and replace them with relevant current references. Any changes to the reference list should be mentioned in the rebuttal letter that accompanies your revised manuscript. If you need to cite a retracted article, indicate the article’s retracted status in the References list and also include a citation and full reference for the retraction notice.

Changes were done in regards to reference list by adding four more articles to broaden the discussion.

° We note that you have indicated that data from this study are available upon request. PLOS only allows data to be available upon request if there are legal or ethical restrictions on sharing data publicly. For information on unacceptable data access restrictions, please see http://journals.plos.org/plosone/s/data-availability#loc-unacceptable-data-access-restrictions. If there are no restrictions, please upload the minimal anonymized data set necessary to replicate your study findings as either Supporting Information files or to a stable, public repository and provide us with the relevant URLs, DOIs, or accession numbers.

There are no restrictions on sharing our dataset and we shared it on public repository recommended by PLOS. 

(URL: https://mfr.osf.io/render?url=https%3A%2F%2Fosf.io%2Fp479v%2Fdownload)

° Thank you for stating the following in the Financial Disclosure section: "The author(s) received no specific funding for this work." We note that one or more of the authors are employed by a commercial company: Zagreb City Pharmacy

a) Please provide an amended Funding Statement declaring this commercial affiliation, as well as a statement regarding the Role of Funders in your study. If the funding organization did not play a role in the study design, data collection and analysis, decision to publish, or preparation of the manuscript and only provided financial support in the form of authors' salaries and/or research materials, please review your statements relating to the author contributions, and ensure you have specifically and accurately indicated the role(s) that these authors had in your study. You can update author roles in the Author Contributions section of the online submission form. Please also include the following statement within your amended Funding Statement.

Zagreb City Pharmacy did not play a role in the study design, data collection and analysis, decision to publish, or preparation of the manuscript and only provided financial support in the form of authors' regular salaries. This sentence was included within our Funding Statement: “The funder provided support in the form of salaries for authors TV and AS, but did not have any additional role in the study design, data collection and analysis, decision to publish, or preparation of the manuscript. The specific roles of these authors are articulated in the ‘author contributions’ section.”

° Please also provide an updated Competing Interests Statement declaring this commercial affiliation along with any other relevant declarations relating to employment, consultancy, patents, products in development, or marketed products, etc. 

This commercial affiliation does not alter our adherence to PLOS ONE policies on sharing data and materials.

° Abstract:

- from 2017 to 2020 (1.76, 1.91, 1.91 and 2.01/1000 inhabitant-days, respectively)

suggestion for clarity: increased in 2017, 2018, 2019 and 2020

We included this suggestion for better clarity.

° Azithromycin distribution to pharmacies was the largest in March 2020 which indicates that fear of a great epidemic can influence consumption of antimicrobials with unproven benefits for the disease in question. Our data indicate that azithromycin overuse was also present during the second wave of the COVID-19 epidemic in Croatia.

I think the DOT over the years is the best measure of AZ overconsumption, as the monthly DOT are difficult to interpret. Distribution is, as the authors correctly state in the limitations, not the same as use; and we can assume that with the quick succession of scientific reports on this issue, it will have been very difficult to attune supply and demand. While I agree that the monthly evolution is described in the abstract, I would personally leave it out of the conclusion.

We included this sentence: Our data showed increased azithromycin distribution in period 2017 – 2020 which indicates azithromycin overuse. Please, see abstract for further details.

° I think it is rather bold to assume that fear has been the main drive for azithromycin consumption. Also, until October, there were to my knowledge still only two randomized controlled trials (Cavalcanti et al, Furtado et al), in which azithromycin was mostly combined with hydroxychloroquine (overview doi: 10.1136/bmjresp-2020-000806). I agree that AZ should have ideally only been prescribed in a context of randomized controlled trials. However, with the conflicting data (and no direct safety issues reported with AZ monotherapy) I can understand why hospitals with more limited resources (in terms of dedicated study personnel) would have chosen to use AZ outside of studies until the results of RECOVERY were widely known. I think this sentence “ Preliminary information on COVID-19 treatments with a desire to offer and try what is available even in the absence of strong scientific evidence might have influenced practices of antimicrobial prescriptions.” Is much stronger, closer to the truth and suits the conclusion better. I would rather reform the current sentence in the conclusion as a question for the introduction, e.g. “could fear of a great epidemic influence consumption of antimicrobials with unproven benefits?”, or else raise this issue in the discussion.

We included this sentence in the conclusion: Preliminary information on COVID-19 treatments with a desire to offer and try what is available even in the absence of strong scientific evidence might have influenced practices of antimicrobial prescriptions.

° About the 8.1% increase compared to the mean of 2017-2019

o Compared to 2018-2019 the increase is only 5%. I have no view on AZ prescription in Croatia, but I feel their case would be stronger if AZ prescription over the last 5 years was mentioned instead of only the last 3. Given that this makes the figure more complex, I would understand why the authors would stick to 3 years, but I would kindly ask if they could provide me those numbers then.

We do not have this data as our study included period 2017 to 2020. However, we included routinely collected data on antibiotics use by Agency for medicinal product and medical devices in Croatia and highlighted the use of azithromycin and its impact in discussion. However, this data is expressed as defined doses (DDD)/1000 inhabitants per day and could not be compared to data we provided in this study.

° I presume that in Croatia too, a significant part of AZ prescription will be chronic AZ use (e.g. 3 weekly doses for COPD), which of course greatly influences the DOT/1000 inhabitant days. I wonder if the average reader will be aware of the fact that an 8,1% increase is therefore more important than it looks at first sight, because treatment duration for COVID is shorter. An absolute increase of 0.15DOT/1000inhabitant days is quite abstract, but in a population of approximately 4000000 inhabitants, and given that a normal AZ treatment course for COVID would be 5 days, I’d say that about an extra 43000 patients (!!) were treated with AZ. I would even suspect that the true overconsumption due to COVID has been larger. One should subtract from the mean of 2017-2019 also the number of cases that have not been treated from atypical pneumonia in the Influenza season, which was considerably less than previous years due to isolation measures. The latter is marginally mentioned in the manuscript but could be emphasized more to underline the magnitude of the increase.

Thank you for raising this interesting point. We agree that the relative number increase is not intuitive but since it includes the population size, we believe it is useful and reports a figure that can be compared to other settings. However, we did include in our revision the absolute number of increases. Actually, the 0.15/1000 DOT increase would correspond to 37224 more 5-day courses of AZ prescriptions in 2020. Yes, we also agree that this might be an underestimation, because of fewer AZ prescriptions for atypical pneumonia in the COVID-19 season. We included a statement on this in the discussion.

 °AZ monotherapy seems safe from the available data. However, combination with HQ or with other antibiotics on ICU raises concerns for cardiac adverse events and warrants close monitoring. This could be emphasized in the discussion and related to the significant increase in (often frail) patients that have been exposed to the drug.

We added two sentences in the discussion on this point: While widespread use of antimicrobials leads to more bacterial resistance which is of great public health concern, it also exposes patients to possible side effects. Azithromycin is generally a well tolerated and safe drug, however, when combined with other QT-prolonging drugs (such as hydroxychloroquine) it may increase the risk for serious toxicities and requires careful monitoring. 

°At the other hand, for the authors, the most important concern on the long-term is probably microbial resistance against macrolides. For this I personally think that it is reassuring that the relative increase has ONLY been 8%. I think that the discussion would be stronger if the authors could elaborate on that.

o E.g. do they think that this 8% increase in DOT, which will have mostly been spread over short treatment courses (causing less resistance than chronic treatment) of approximately 5 days, will have a significant impact on bacterial resistance? To the best of my knowledge, there is no data for my own country yet… is there for Croatia?

We justified our fear of further macrolide resistance in discussion. We included this in the discussion: Even though an increase of 8.1% of total azithromycin DOT from the average of 2017¬¬¬–2019 compared to 2020 seems modest, we should highlight the fact that Croatia is already one of the countries with highest percentage of macrolide resistant isolates among EU/EEA countries [20]. Macrolides were the second most common used antibiotics with azithromycin accounting for most of the use in Croatia in 2014 – 2019 with daily defined doses (DDD)/1000 inhabitants per day of 2.58, 2.82, 2.55, 2.56, 2.73, 2.72, respectively, after penicillins including beta-lactamase inhibitors [21,22]. Sample of Streptococcus pneumoniae with estimated national population coverage of 80% showed increasing trend in macrolides resistance in the period 2015 – 2018 (19, 34, 36, 32%, respectively) [20]. Resistance data collected from 38 centers in Croatia in 2019 showed 31% resistant S. pneumoniae to macrolides with a range of local results from 0 to 53% and 9% of macrolide resistant Streptococcus pyogenes with a range of local results from 0 to 25% [23].

---

## [Decision Letter · Decision Letter 1]

28 Jul 2021

PONE-D-21-11999R1

Azithromycin consumption during the COVID-19 pandemic in Croatia, 2020

PLOS ONE

Dear Dr. Bogdanic,

Thank you for submitting your manuscript to PLOS ONE. After careful consideration, we feel that it has merit but does not fully meet PLOS ONE’s publication criteria as it currently stands. Therefore, we invite you to submit a revised version of the manuscript that addresses the points raised during the review process.

ACADEMIC EDITOR: Please check the reviewers comments...

We look forward to receiving your revised manuscript.

Kind regards,

Prasenjit Mitra, MD, MRSB, MIScT, FLS, FACSc, FAACC

Academic Editor

PLOS ONE

Journal Requirements:

Reviewers' comments:

Reviewer's Responses to Questions

**Comments to the Author**

1. If the authors have adequately addressed your comments raised in a previous round of review and you feel that this manuscript is now acceptable for publication, you may indicate that here to bypass the “Comments to the Author” section, enter your conflict of interest statement in the “Confidential to Editor” section, and submit your "Accept" recommendation.

Reviewer #1: All comments have been addressed

2. Is the manuscript technically sound, and do the data support the conclusions?

Reviewer #1: Yes

3. Has the statistical analysis been performed appropriately and rigorously? 

Reviewer #1: Yes

4. Have the authors made all data underlying the findings in their manuscript fully available?

Reviewer #1: Yes

5. Is the manuscript presented in an intelligible fashion and written in standard English?

Reviewer #1: Yes

6. Review Comments to the Author

Reviewer #1: Again I compliment the authors with the well written article, providing insight in azithromycin use, something many suspect but few have published upon. I have some minor suggestions left. No need for re-review after this.

Best wishes

IG

Abstract:

I would change the order of sentences to draw first attention to the real solid conclusion of the manuscript. Also maybe better “the period”. Also I’d dare to use may instead of might (stronger).

=> Our data showed increased azithromycin distribution in the period 2017 – 2020 which indicates azithromycin overuse. Preliminary information on COVID-19 treatments with a desire to offer and try what is available even in the absence of strong scientific evidence may have influenced practices of antimicrobial prescriptions.

Discussion:

* Galotti et al. analyzed Twitter messages posted during the early stages of the COVID-19 pandemic and found that potentially unreliable information preceded the rise of COVID-19 cases [8].

I’m not sure that I get this sentence. The way I interpret it, is that in the early pandemic, there was a lot of misinformation. The way it is formulated though, one gets the impression that the rise of COVID-19 cases has anything to do with the previous misinformation. To avoid any confusion, I’d maybe say:

=> Galotti et al. analyzed Twitter messages posted from January to March 2020, and found an increase in potentially unreliable information in those early stages of the COVID-19 pandemic [8].

* This indicates that fear of a great epidemic and uncertainty due to the absence of the treatment with proven efficacy can influence consumption of antimicrobials with assumed or unproven benefits.

Still think “indicates” is somewhat strong based only on the associations given before. If I may add a couple of suggestions:

=> This supports the idea/This supports the thesis/It thus seems… that fear of a great epidemic and uncertainty due to the absence of the treatment with proven efficacy can influence consumption of antimicrobials with assumed or unproven benefits.

* Isolation measures and social distancing during 2020 probably contributed to fewer acute respiratory infections including influenza and atypical pneumonia which is the common reason for azithromycin use during the influenza season.

=> Maybe “post-influenza pneumococcal infection” instead of influenza?

7. PLOS authors have the option to publish the peer review history of their article (what does this mean?). If published, this will include your full peer review and any attached files.

Reviewer #1: **Yes: **Iwein Gyselinck

---

## [Author Response · Author response to Decision Letter 1]

31 Jul 2021

° Reviewer #1: Again I compliment the authors with the well written article, providing insight in azithromycin use, something many suspect but few have published upon. I have some minor suggestions left. No need for re-review after this.

We are very grateful for Reviewer’s compliments and all suggestions that improved the quality of our manuscript.

° Abstract:

I would change the order of sentences to draw first attention to the real solid conclusion of the manuscript. Also maybe better “the period”. Also I’d dare to use may instead of might (stronger).

=> Our data showed increased azithromycin distribution in the period 2017 – 2020 which indicates azithromycin overuse. Preliminary information on COVID-19 treatments with a desire to offer and try what is available even in the absence of strong scientific evidence may have influenced practices of antimicrobial prescriptions.

We replaced the current conclusion with this suggestion.

° Discussion:

* Galotti et al. analyzed Twitter messages posted during the early stages of the COVID-19 pandemic and found that potentially unreliable information preceded the rise of COVID-19 cases [8].

I’m not sure that I get this sentence. The way I interpret it, is that in the early pandemic, there was a lot of misinformation. The way it is formulated though, one gets the impression that the rise of COVID-19 cases has anything to do with the previous misinformation. To avoid any confusion, I’d maybe say:

=> Galotti et al. analyzed Twitter messages posted from January to March 2020, and found an increase in potentially unreliable information in those early stages of the COVID-19 pandemic [8].

We replaced the current sentence with this suggestion.

°* This indicates that fear of a great epidemic and uncertainty due to the absence of the treatment with proven efficacy can influence consumption of antimicrobials with assumed or unproven benefits.

Still think “indicates” is somewhat strong based only on the associations given before. If I may add a couple of suggestions:

=> This supports the idea/This supports the thesis/It thus seems… that fear of a great epidemic and uncertainty due to the absence of the treatment with proven efficacy can influence consumption of antimicrobials with assumed or unproven benefits.

We used this suggestion (It thus seems…).

°* Isolation measures and social distancing during 2020 probably contributed to fewer acute respiratory infections including influenza and atypical pneumonia which is the common reason for azithromycin use during the influenza season.

=> Maybe “post-influenza pneumococcal infection” instead of influenza?

Ok, we included this.

---

## [Decision Letter · Decision Letter 2]

23 Sep 2021

PONE-D-21-11999R2Azithromycin consumption during the COVID-19 pandemic in Croatia, 2020PLOS ONE

Dear Dr. Bogdanic,

Thank you for submitting your manuscript to PLOS ONE. After careful consideration, we feel that it has merit but does not fully meet PLOS ONE’s publication criteria as it currently stands. Therefore, we invite you to submit a revised version of the manuscript that addresses the points raised during the review process.

ACADEMIC EDITOR: Please look into the reviewer's comments

We look forward to receiving your revised manuscript.

Kind regards,

Prasenjit Mitra, MD, CBiol, MRSB, MIScT, FLS, FACSc, FAACC

Academic Editor

PLOS ONE

Journal Requirements:

Additional Editor Comments (if provided):

Reviewers' comments:

Reviewer's Responses to Questions

**Comments to the Author**

1. If the authors have adequately addressed your comments raised in a previous round of review and you feel that this manuscript is now acceptable for publication, you may indicate that here to bypass the “Comments to the Author” section, enter your conflict of interest statement in the “Confidential to Editor” section, and submit your "Accept" recommendation.

Reviewer #1: All comments have been addressed

2. Is the manuscript technically sound, and do the data support the conclusions?

Reviewer #1: (No Response)

3. Has the statistical analysis been performed appropriately and rigorously? 

Reviewer #1: Yes

4. Have the authors made all data underlying the findings in their manuscript fully available?

Reviewer #1: Yes

5. Is the manuscript presented in an intelligible fashion and written in standard English?

Reviewer #1: Yes

6. Review Comments to the Author

Reviewer #1: - ... fewer acute respiratory infections including post-influenza pneumococcal infection and atypical pneumonia which is THE common reason for azithromycin use during the influenza season ...: use A common reason instead of THE

- ... So, the facts that there was no significant influenCa epidemic would indicate that the increase in azithromycin...: typo influenZa

7. PLOS authors have the option to publish the peer review history of their article (what does this mean?). If published, this will include your full peer review and any attached files.

Reviewer #1: **Yes: **Iwein Gyselinck

---

## [Author Response · Author response to Decision Letter 2]

23 Sep 2021

We are grateful to the reviewer again for the comments on our paper. We have incorporated suggestions and made necessary changes. Enclosed herein is the revised version of our manuscript. Please see below, in italics, for a response to the reviewer’s comments. 

° fewer acute respiratory infections including post-influenza pneumococcal infection and atypical pneumonia which is THE common reason for azithromycin use during the influenza season ...: use A common reason instead of THE

We changed this in the manuscript. 

° So, the facts that there was no significant influenCa epidemic would indicate that the increase in azithromycin...: typo influenZa

We corrected this typo.

Nikolina Bogdanić

---

## [Decision Letter · Decision Letter 3]

21 Oct 2021

PONE-D-21-11999R3

Azithromycin consumption during the COVID-19 pandemic in Croatia, 2020

PLOS ONE

Dear Dr. Bogdanic,

Thank you for submitting your manuscript to PLOS ONE. After careful consideration, we have decided that your manuscript does not meet our criteria for publication and must therefore be rejected.

I am sorry that we cannot be more positive on this occasion, but hope that you appreciate the reasons for this decision.

Yours sincerely,

Farzad Taghizadeh-Hesary

Academic Editor

PLOS ONE

Editor Comments:

Summary: To report the rate of azithromycin administration over the pandemic and to find a correlation with Covid-19 cases.

Overall scoring (1-5)

1. Novelty: 3

2. Writing: 4

3. Methodology: 2

4. Data presentation: 3

5. Data interpretation: 3

Major concerns:

- Statistical analysis: A investigator needs to examine three assumptions to run Spearman's Rank-Order Correlation test.

1. Type of variables: ordinal, interval, or ratio scale

2. Paired observation

3. Monotonic relationship

In this study, the 2nd assumption is not considered. And, the 3rd one is not addressed.

- This study has tried to rule out the effect of other factors influencing the rate of macrolide consumption by referring to studies of other countries (for example, coinfection to a study from North California, or Influenza to a general report from Europe). To make the results more reliable, the investigators need to evaluate and rule out the other reasons for azithromycin overuse during the pandemic IN CROATIA.

Comments:

- it is suggested to improve the clinical implications of the study with the following two suggestions:

1. to add the information of azithromycin adverse effects during the pandemic.

2. to compare the covid-19 mortality based on a rate of azithromycin administration.

Reviewers' comments:

Reviewer's Responses to Questions

**Comments to the Author**

1. If the authors have adequately addressed your comments raised in a previous round of review and you feel that this manuscript is now acceptable for publication, you may indicate that here to bypass the “Comments to the Author” section, enter your conflict of interest statement in the “Confidential to Editor” section, and submit your "Accept" recommendation.

Reviewer #1: All comments have been addressed

2. Is the manuscript technically sound, and do the data support the conclusions?

Reviewer #1: Yes

3. Has the statistical analysis been performed appropriately and rigorously? 

Reviewer #1: Yes

4. Have the authors made all data underlying the findings in their manuscript fully available?

Reviewer #1: Yes

5. Is the manuscript presented in an intelligible fashion and written in standard English?

Reviewer #1: Yes

6. Review Comments to the Author

Reviewer #1: (No Response)

7. PLOS authors have the option to publish the peer review history of their article (what does this mean?). If published, this will include your full peer review and any attached files.

Reviewer #1: **Yes: **Iwein Gyselinck, MD

Clinical Resident Respiratory Diseases, (subgroup), UZ Leuven

PhD student BREATHE, department CHROMETA, KU Leuven

Campus Gasthuisberg - O&N1bis Herestraat 49 bus 706

B-3000 Leuven, Belgium

- - - - -

---

## [Author Response · Author response to Decision Letter 3]

12 Nov 2021

We wish to appeal the decision on the manuscript „Azithromycin consumption during the COVID-19 pandemic in Croatia”, 2020 PONE-D-21-11999R3. 

A point-by-point response to the Editors’ comments, which are repeated in italics, is given below. 

**Overall scoring (1-5)

1. Novelty: 3

2. Writing: 4

3. Methodology: 2

4. Data presentation: 3

5. Data interpretation: 3 

We strongly disagree with the judgement that our manuscript is not methodologically sound. The statistical analysis in our manuscript is simple and follows the usual presentation of such data published in other studies. Namely the main focus of the analysis was the comparison of drug consumption (in this case azithromycin) in a per year or monthly interval in one year (2020 in this case) to the consumption of the same drug in the same period to previous years (2017, 2018, 2019). Our main analysis (see the first paragraph of the results section) was focused on the increase of azithromycin consumption expressed in percentages (“This was a 5.2% increase from 2019 to 2020 and an 8.1% increase from the average of 2017–2019 to 2020. This 8.1% increase corresponds to a total of 37224 5-day courses of azithromycin prescriptions.”). We also focused on how many times more azithromycin was prescribed (see abstract: “Non-hospital pharmacies received 2.18 times and hospital pharmacies 4.39 times more DOT units/1000 inhabitants of azithromycin in March 2020 compared to the average distribution rate in March 2017–2019.") During the peak of the COVID-19 epidemic (November and December 2020) azithromycin distribution increased considerably in hospitals (3.62 and 3.19 times, respectively) and non-hospital pharmacies (1.93 and 1.84 times, respectively) compared to the average consumption in the same months in 2017–2019.)” We also presented our data in a figure and the monthly distribution of the consumption of azithromycin in 2020 and was very different than in years 2017-2019 (see figure). So, our main finding does not depend on the Spearman correlation analysis.

** Statistical analysis: A investigator needs to examine three assumptions to run

Spearman's Rank-Order Correlation test.

1. Type of variables: ordinal, interval, or ratio scale

2. Paired observation

3. Monotonic relationship

In this study, the 2nd assumption is not considered. And, the 3rd one is not addressed.

In the context of the monthly distribution from July to December 2020 we compared the absolute number of COVID-19 cases from July to December 2020 with the total azithromycin consumption (the definition of consumption in clearly stated in the method section “Azithromycin consumption was measured as days of therapy (DOT) which represents a ratio of the total dose of azithromycin in one package divided by the defined daily dose unit of 0.3 gr.”). By examining the figure it can be concluded that all assumptions for the Spearman correlation coefficient were met (this seems to be clear to the first reviewer). The variables are continuous. The data is paired by months (the number of COVID-19 cases in July are paired with the consumption of azithromycin in July, and the same is paired for August, September, October, November and December), so this is a paired sample. The figures also show that the increase in COVID-19 cases is correlated with the increase in azithromycin consumption in the period from July to December 2020. Hence the assumption of monotonic relationship is met (“In a monotonic relationship, the variables tend to move in the same relative direction, but not necessarily at a constant rate”… “Linear relationships are also monotonic” see for example https://support.minitab.com/en-us/minitab/19/help-and-how-to/statistics/basic-statistics/supporting-topics/basics/linear-nonlinear-and-monotonic-relationships/). Enclosed to our manuscript as S1 Supporting Information are graph (Figure A) and table with original values (Table A) further supporting the monotonic relationship of COVID-19 cases and azithromycin consumption. So, all requirements to perform Spearman’s correlation are met. 

**- This study has tried to rule out the effect of other factors influencing the rate of macrolide consumption by referring to studies of other countries (for example, coinfection to a study from North California, or Influenza to a general report from Europe). To make the results more reliable, the investigators need to evaluate and rule out the other reasons for azithromycin overuse during the pandemic IN CROATIA.

Our analysis was not around “ruling out other factors influencing the rate of macrolide consumption”. We simply stated in our discussion that we did not have an influenza epidemic during the winter months of 2020 and that coinfection with other pathogens in COVID-19 is rare. It is well known that during the influence epidemic there is an increase of antibiotic consumption. We provided data of azithromycin distribution in Croatia for the previous three years (2017-2019) and it showed that the monthly pattern of azithromycin distribution was similar in 2017, 2018 and 2019 and completely different in the COVID-19 year (2020). The suggestion of the editorial review that there might have been other causes (other than COVID-19) of increase azithromycin consumption in Croatia in 2020 compared to previous 3-years is very unlikely and this has been now discussed in our paper. Furthermore, the three authors (NB, LM, JB) of the manuscript work at the only special 220-bed (exclusively) infectious diseases hospital in the capital of Croatia, Zagreb. At the beginning of the COVID-19 epidemic in Croatia (March 2020), there was a decision of the management of the hospital to purchase large quantities of azithromycin for treatment of COVID-19 because of speculations that azithromycin might help in the treatment of COVID-19. At that time and afterwards in 2020 there was no other infectious disease epidemic going on in Zagreb nor in Croatia that would need such large quantities of azithromycin for treatment. This (azithromycin overconsumption) seems to have happened in other countries and settings, as the first reviewer alluded to (Dr Iwein Gyselinck stated “I compliment the authors with the well written article, providing insight in azithromycin use, something many suspect but few have published upon”).

Moreover, healthcare professionals in Croatia are obligated to report 92 different infectious disease entities including all forms of pneumonia (including those caused by M. pneumoniae, Chlamydia, etc; document in Croatian: https://narodne-novine.nn.hr/clanci/sluzbeni/1994_03_23_410.html) to the Croatian Institute of Public Health. There was no report from the Croatian National Public Health Institute of a large number of non-Covid pneumonias occurring in the whole 2020. It would be highly unlikely that during the outbreak of COVID-19 we simultaneously had a large outbreak of another infectious disease that is treated by azithromycin. We added two sentences to our original manuscript in the discussion section: “One may also speculate about other non-COVID-19 causes of azithromycin overconsumption in 2020. However, this is very unlikely as there was no report from the Croatian National Public Health Institute of any other ongoing epidemic requiring azithromycin treatment in 2020.”

We also received two suggestions from the editor: 1. to add the information of azithromycin adverse effects during the pandemic and 2. to compare the covid-19 mortality based on a rate of azithromycin administration. However, we did not analyse individual data so it is impossible to address those two issues in a meaningful way. Also, the aim of our study was not to assess factors related to COVID-19 mortality, so this analysis on the data we have would not give a valid result.

 Finally, as we consider this topic needs prompt attention of healthcare workers who have a role in treatment of COVID-19 patients we posted our manuscript on preprint server medRxiv on 1st of November 2021: https://medrxiv.org/cgi/content/short/2021.10.31.21265714v1.

Thank you for giving us the opportunity to respond. We look forward to hearing from you in a reasonable time. We would be glad to respond to any further questions and comments that you may have.

On behalf of all authors,

Nikolina Bogdanić and Josip Begovac

---

## [Decision Letter · Decision Letter 4]

28 Dec 2021

PONE-D-21-11999R4Azithromycin consumption during the COVID-19 pandemic in Croatia, 2020PLOS ONE

Dear Dr. Bogdanic,

Thank you for submitting your manuscript to PLOS ONE. After careful consideration, we feel that it has merit but does not fully meet PLOS ONE’s publication criteria as it currently stands. Therefore, we invite you to submit a revised version of the manuscript that addresses the points raised during the review process.

Please revise the statistical application and data presentation as recommended by the reviewer. Please see the comments directly on the manuscript. 

We look forward to receiving your revised manuscript.

Kind regards,

Iddya Karunasagar

Academic Editor

PLOS ONE

Journal Requirements:

Additional Editor Comments (if provided):

Please see the comments of the expert on statistics. Please apply statistics as recommended by the reviewer and modify the presentation of data.

Reviewers' comments:

Reviewer's Responses to Questions

**Comments to the Author**

1. If the authors have adequately addressed your comments raised in a previous round of review and you feel that this manuscript is now acceptable for publication, you may indicate that here to bypass the “Comments to the Author” section, enter your conflict of interest statement in the “Confidential to Editor” section, and submit your "Accept" recommendation.

Reviewer #2: All comments have been addressed

Reviewer #3: All comments have been addressed

2. Is the manuscript technically sound, and do the data support the conclusions?

Reviewer #2: Yes

Reviewer #3: Partly

3. Has the statistical analysis been performed appropriately and rigorously? 

Reviewer #2: No

Reviewer #3: I Don't Know

4. Have the authors made all data underlying the findings in their manuscript fully available?

Reviewer #2: Yes

Reviewer #3: (No Response)

5. Is the manuscript presented in an intelligible fashion and written in standard English?

Reviewer #2: Yes

Reviewer #3: (No Response)

6. Review Comments to the Author

Reviewer #2: Comments

• To study the correlation authors can modify the plot by taking independent variable (No of Covid -19 cases) on X- axis and dependent variable (Azithromycin usage) on Y- axis as per below (reference plot). Exclude the monthly distribution data for this analysis. Graph representing monthly distribution can still be retained and explained separately.

Spearman correlation is employed when two variables are monotonically related. Which means all the data should be entirely in an increasing trend or in a decreasing trend and not a combination of both. The present data doesn’t strictly follow a monotonous relation neither a linear relation (Azithromycin usage in the month of march is an outlier)

Authors can explore qualities of both correlation test (Pearson’s product moment and spearman’s rank correlation) and write the discussion accordingly

For instance, spearman’s correlation explains the strength of monotonous relation, so the explanation for this could be how strict the data was in terms of monotonous relation (every increase in Covid-19 case resulting in increase of azithromycin usage)

Pearson’s correlation explains strength of linear correlation, so the explanation for this could be if the rate of increase in the Azithromycin usage was constant for every increase in Covid-19 case

• Comparison between monthly usage of Azithromycin for the year 2020 and the average use of azithromycin for the period 2017-2019 could be tested for significance. Authors can use ANOVA pair wise comparison and rate the difference as per p-value which can be denoted in graph using asterisks

Reviewer #3: The article looks interesting, not very hard earned statistical paper but good evidence is present. the study is not a part of dual publication. the concern in the science is good enough

7. PLOS authors have the option to publish the peer review history of their article (what does this mean?). If published, this will include your full peer review and any attached files.

Reviewer #2: **Yes: **Caroline DSouza

Reviewer #3: No

---

## [Author Response · Author response to Decision Letter 4]

7 Jan 2022

We are grateful to the reviewers for the comments on our paper. Enclosed herein is the revised version of our manuscript with supplemental material (S1 Supporting Information). A point-by-point response to the Reviewers’ comments, which are repeated in italics, is given below.

Firstly, we would like to emphasize that the main focus of the analysis in our study was to describe the distribution of azithromycin to the hospital and non-hospital pharmacies in the years 2017 to 2020. Azithromycin distribution per months to non-hospital and hospital pharmacies was planned to be presented graphically. Our main analysis was focused on the difference in azithromycin consumption expressed in percentages. So, our main findings do not depend on the correlation analysis. 

Reviewer #2: Comments

• To study the correlation authors can modify the plot by taking independent variable (No of Covid -19 cases) on X- axis and dependent variable (Azithromycin usage) on Y- axis as per below (reference plot). Exclude the monthly distribution data for this analysis. Graph representing monthly distribution can still be retained and explained separately. Spearman correlation is employed when two variables are monotonically related. Which means all the data should be entirely in an increasing trend or in a decreasing trend and not a combination of both. The present data doesn’t strictly follow a monotonous relation neither a linear relation (Azithromycin usage in the month of march is an outlier)

Authors can explore qualities of both correlation test (Pearson’s product moment and spearman’s rank correlation) and write the discussion accordingly

For instance, spearman’s correlation explains the strength of monotonous relation, so the explanation for this could be how strict the data was in terms of monotonous relation (every increase in Covid-19 case resulting in increase of azithromycin usage)

Pearson’s correlation explains strength of linear correlation, so the explanation for this could be if the rate of increase in the Azithromycin usage was constant for every increase in Covid-19 case.

Spearman’s rank-order correlation test was used to measure the association between quantities of azithromycin distribution and the number of COVID-19 cases from July to December 2020. There was a positive correlation between the number of COVID-19 cases and the total azithromycin consumption from July to December 2020 (ρ=0.94, p=0.005). July 2020 was the first month we included in this analysis due to the obvious increase in number of COVID-19 cases (531 cases in June, 2361 case in July). 

We are a bit puzzled by the statement of the reviewer “Spearman correlation is employed when two variables are monotonically related. Which means all the data should be entirely in an increasing trend or in a decreasing trend and not a combination of both.” If this means that a monotonic relationship is a prerequisite for performing the Spearman rho we disagree. If this is not the case, we apologize for the explanation on why a monotonic relationship is not a prerequisite for performing the Spearman test. 

We have not been able to confirm that a monotonic relationship is a prerequisite for performing the Spearman test by looking at statistical textbooks (e.g. Armitage, Berry, Matthews: Statistical Methods in Medical Research ed 4 2002. A. Petrie and Sabin C. Medical statistics at a Glance 3rd edition 2009) nor does an online textbook https://statistics.laerd.com/statistical-guides/spearmans-rank-order-correlation-statistical-guide.php mention this. On the contrary, on the Laerd statistics web site we can find the following sentence “A monotonic relationship is not strictly an assumption of Spearman's correlation. That is, you can run a Spearman's correlation on a non-monotonic relationship to determine if there is a monotonic component to the association.” The question on the “monotonic assumption” of Spearman’s test appeared at least 3 times in the statsexchange forum (see https://stats.stackexchange.com/questions/109793/spearmans-monotonic ; https://stats.stackexchange.com/questions/161376/definition-of-monotonic-and-implications-for-using-spearmans-correlation ; https://stats.stackexchange.com/questions/149350/how-to-identify-that-relationship-between-two-variables-are-monotonic-or-not). All responses mentioned that a monotonic relationship is not a prerequisite for performing Spearman’s test because the test is testing whether a monotonic relationship is present or not or how strong it is. One of the commentators mentioned that it would be “strange and not true” to “test only for a monotone relationship if you already know there's a monotone relationship." This question also came up at Research Gate: (https://www.researchgate.net/post/Can_I_run_spearmans_correlation_on_non_monotonic_data_Is_it_advisable) and again the answer of several commentators was in the line: “Yes, there is nothing wrong with using Spearman correlation coefficient” (on non-monotonic data). Specifically, the quoted comment was provided by Abolfazl Ghoodjani, who is presently professor of statistics at the Mc Gill University, Montreal Canada. 

As suggested by Reviewer and Editor, we included additional analyses as supplemental material (S1 Supporting Information) and included comments about it in the Methods and Discussion. We provided a graph on the Spearman-rho which was calculated in our manuscript. We also provided additional statistical analysis but would like to stress again that this study was not planned as a hypothesis-driven study. We simply wanted to present descriptive data on azithromycin distribution between 2017 to 2020. We present our main data in absolute numbers of DOTs based on 1000 inhabitants-days of azithromycin distribution. The comparisons between years are reported as percent or rate difference. The monthly DOTs are reported per 1000 inhabitants and presented graphically. Since this seems not to be clear we rewrote the Methods section and included what our main analysis was about. We believe that the Spearman rho and the additional statistics should not be the main focus of our study and they contribute little to our main conclusion. This is why we put them into the Supplemental material.

• Comparison between monthly usage of Azithromycin for the year 2020 and the average use of azithromycin for the period 2017-2019 could be tested for significance. Authors can use ANOVA pair wise comparison and rate the difference as per p-value which can be denoted in graph using asterisks

Thank you for this comment. The additional statistics includes a t-test because we compare only two groups (one way ANOVA and t-test are equivalent with 2 groups). We would like to point out that we do not have individual (daily or weekly data). We have only total DOTs for each month for the period 2017 to 2020. So, for example, we cannot compare by ANOVA January 2020 to anything else because we have only one number for January 2020. We included into the revised manuscript a comparison of the mean monthly dispension of azithromycin by quarters. In this analysis the number of observations in the mean monthly quarterly comparison is small (e.g. 3 in the group 2020 versus 3 in the group 2017-2019), so the test has little power to detect a difference. Therefore, it is not surprising that we do not have a significant P in the 1st and 4th quarter. Also, there is no significant difference in the monthly distribution for all 12 months when 2020 is compared to the average of 2017 to 2019 (and yes the variances are not equal). But does this formal statistical analysis of the mean monthly comparison for the whole year make sense if you look at the pattern of monthly distribution (see graphs)? Also, the nonsignificant P-values derived from the above-mentioned analyses should not be interpreted as absence of overconsumption of azithromycin in 2020. Almost 38 000 extra 5-day courses of azithromycin were distributed in 2020 compared with previous years, with no obvious other reasons but COVID-19 which is quite relevant despite a „nonsignficant“ P. Of note, COVID-19, as discussed in our paper, should not be treated with azithromycin. Let’s not loose common sense when interpreting data. 

Reviewer #3: The article looks interesting, not very hard earned statistical paper but good evidence is present. the study is not a part of dual publication. the concern in the science is good enough

We thank for this comment. We don't think that every paper need hypothesis-driven inferential statistics if enough evidence is present to draw conclusions from the description of the data. Moreover, similar studies published in high-quality journals did not report any P-values and were based on description (for example: King LM et al. Trends in U.S. outpatient antibiotic prescriptions during the COVID-19 pandemic. Clin Infect Dis. 2020 Dec 29. doi:10.1093/cid/ciaa1896). We agree with the reviewer that “good evidence is present”.

We consider this topic needs prompt attention of healthcare workers who have a role in treatment of COVID-19 patients to consider more careful antimicrobial prescribing. We appreciate the valuable time of all Reviewers and Editors and also all suggestions they had so far but we must say that the period of almost 9 months for evaluation of this manuscript is quite long. Our original submission was rejected based on the wrong judgment of the Editor in charge that the Spearman test was inappropriately applied to our data. Please note there is nothing wrong with the Spearman test in our analysis and, in addition, the main result of our study does not even depend on the result of Spearman’s rho. The other reviewer of our original submission (Dr Iwein Gyselinck), similarly to the one of the current reviewers, also concluded that the methods used for reaching our conclusion are appropriate and sound and had no issues regarding the use of Spearmans test. Dr Iwein Gyselinck, an expert in the field of respiratory diseases, recommended publishing our paper.

We thank you for your reevaluation of our manuscript and consideration for publication. 

On behalf of all authors,

Nikolina Bogdanić and Josip Begovac

---

## [Decision Letter · Decision Letter 5]

20 Jan 2022

Azithromycin consumption during the COVID-19 pandemic in Croatia, 2020

PONE-D-21-11999R5

Dear Dr. Bogdanic,

We’re pleased to inform you that your manuscript has been judged scientifically suitable for publication and will be formally accepted for publication once it meets all outstanding technical requirements.

Kind regards,

Iddya Karunasagar

Academic Editor

PLOS ONE

Additional Editor Comments (optional):

Though technically the manuscript is now in an acceptable form, the authors may note the reviewers comments on the questions raised and answers provided by the authors.

Reviewers' comments:

Reviewer's Responses to Questions

**Comments to the Author**

1. If the authors have adequately addressed your comments raised in a previous round of review and you feel that this manuscript is now acceptable for publication, you may indicate that here to bypass the “Comments to the Author” section, enter your conflict of interest statement in the “Confidential to Editor” section, and submit your "Accept" recommendation.

Reviewer #2: All comments have been addressed

2. Is the manuscript technically sound, and do the data support the conclusions?

Reviewer #2: Yes

3. Has the statistical analysis been performed appropriately and rigorously? 

Reviewer #2: Yes

4. Have the authors made all data underlying the findings in their manuscript fully available?

Reviewer #2: Yes

5. Is the manuscript presented in an intelligible fashion and written in standard English?

Reviewer #2: Yes

6. Review Comments to the Author

Reviewer #2: • Authors have provided long explanation to justify ‘monotonous relation is not a prerequisite’ which was totally unnecessary

It was clearly mentioned to authors that they can explore the qualities of both tests. Spearman correlation explains the strength of monotonous relation” (written in the comment) thus authors were asked to write the discussion accordingly. Also, in the previous comments it was clearly mentioned what monotonic relation means in the context of analysis done by the authors.

• “you can run a Spearman's correlation on a non-monotonic relationship to determine if there is a monotonic component to the association “

Explanation provided by the authors is clearly aligned with review given earlier. So, the whole lot of explanation in this aspect doesn’t serve the purpose for which it was asked

• We also provided additional statistical analysis but would like to stress again that this study was not planned as a hypothesis-driven study. We simply wanted to present descriptive data on azithromycin distribution between 2017 to 2020. We present our main data in absolute numbers of DOTs based on 1000 inhabitants-days of azithromycin Distribution

For the fact that it is mentioned by the authors “There was a positive correlation between the number of COVID-19 cases and the total azithromycin consumption from July to December 2020 (Spearman’s test, ρ=0.94, p=0.005).” authors were asked to produce a statistical evidence and nowhere it was asked to make this as the main focus. All the conclusions drawn from the study also highlights azithromycin over-usage during COVID -19. Rp =Pearson’s product moment correlation coefficient, Rs = Spearman’s rank correlation coefficient can be added in the footnote.

• Authors state “But does this formal statistical analysis of the mean monthly comparison for the whole year make sense if you look at the pattern of monthly distribution (see graphs)?” in contrary authors explain “Among non-hospital pharmacies, azithromycin distribution in March 2020 was 2.18 times higher” “1.93 and 1.84 times higher in November and December 2020” “In April-August 2020 the distributed DOT of azithromycin was not only lower than in other months of 2020, but also lower than the average in April-August 2017–2019 for both hospital and non-hospital pharmacies”” In fact, the distributed DOT of azithromycin in April-June 2020 was statistically significantly lower compared to the average of 2017–2019 (Table A in the S1 supporting information)”

Authors are not asked to provide pooled data (whole year or quarterly) which becomes nonsignificant because of the large variation and outliers. Since there is no daily data on azithromycin usage (reported by the authors presently) test of significance for comparison between the months can avoided.

• Also, the nonsignificant P-values derived from the above-mentioned analyses should not be interpreted as absence of overconsumption of azithromycin in 2020. Almost 38 000 extra 5-day courses of azithromycin were distributed in 2020 compared with previous years, with no obvious other reasons but COVID-19 which is quite relevant despite „nonsignficant“ P.

Authors have done some analysis by grouping the data on their own and explained why it is not relevant which was not asked anywhere.

• Of note, COVID-19, as discussed in our paper, should not be treated with azithromycin. Let’s not loose common sense when interpreting data.

Correlational test with significant r value doesn’t show a causal relation and rather it only shows association. Authors can use appropriate language while answering the comments. Of note, Covid-19 even otherwise whether concluded in this paper or not, is to be treated with antivirals and not antibiotics. Authors are asked to read comments carefully before being opinionated. Sticking to the point with scientific facts and research proof would be appreciated.

7. PLOS authors have the option to publish the peer review history of their article (what does this mean?). If published, this will include your full peer review and any attached files.

Reviewer #2: No

---

## [Editor Report · Acceptance letter]

24 Jan 2022

PONE-D-21-11999R5 

Azithromycin consumption during the COVID-19 pandemic in Croatia, 2020 

Dear Dr. Bogdanić:

I'm pleased to inform you that your manuscript has been deemed suitable for publication in PLOS ONE. Congratulations! Your manuscript is now with our production department. 

Kind regards, 

on behalf of

Dr. Iddya Karunasagar 

Academic Editor

PLOS ONE